# Plant virus movement proteins originated from jelly-roll capsid proteins

**Anamarija Butkovic[1], Valerian V. Dolja[2], Eugene V. Koonin[3], Mart Krupovic [1]***

**1** Institut Pasteur, Université Paris Cité, CNRS UMR6047, Archaeal Virology Unit, Paris, France,
**2** Department of Botany and Plant Pathology, Oregon State University, Corvallis, Oregon, United States of America, **3** National Center for Biotechnology Information, National Library of Medicine, Bethesda, Maryland, United States of America

* mart.krupovic@pasteur.fr

## Abstract

Numerous, diverse plant viruses encode movement proteins (MPs) that aid the virus movement through plasmodesmata, the plant intercellular channels. MPs are essential for virus spread and propagation in distal tissues, and several unrelated MPs have been identified. The 30K superfamily of MPs (named after the molecular mass of tobacco mosaic virus MP, the classical model of plant virology) is the largest and most diverse MP variety, represented in 16 virus families, but its evolutionary origin remained obscure. Here, we show that the core structural domain of the 30K MPs is homologous to the jelly-roll domain of the capsid proteins (CPs) of small RNA and DNA viruses, in particular, those infecting plants. The closest similarity was observed between the 30K MPs and the CPs of the viruses in the families *Bromoviridae* and *Geminiviridae*. We hypothesize that the MPs evolved via duplication or horizontal acquisition of the CP gene in a virus that infected an ancestor of vascular plants, followed by neofunctionalization of one of the paralogous CPs, potentially through the acquisition of unique N- and C-terminal regions. During the subsequent coevolution of viruses with diversifying vascular plants, the 30K MP genes underwent explosive horizontal spread among emergent RNA and DNA viruses, likely permitting viruses of insects and fungi that coinfected plants to expand their host ranges, molding the contemporary plant virome.

## Introduction

Viruses are ubiquitous, obligate intracellular parasites that infect (nearly) all life forms and show enormous diversity with respect to the routes of genome replication and expression, genome size, and gene composition [1–5]. This immense variability notwithstanding, virus proteins can be divided into 3 broad classes involved in distinct functions: (1) genome replication and expression; (2) virion assembly and structure; and (3) virus–host interactions [6,7]. The evolutionary trajectories of the proteins in the first 2 classes are drastically different from those in the third class. Proteins involved in replication and virion structure formation apparently were captured by viruses from hosts at early stages of evolution, and certain replication system components might even descend from primordial replicators antedating the emergence of modern type cells. Some of these proteins are virus hallmarks shared by many groups of

**Data Availability Statement:** All protein sequences in this study can be downloaded from GenBank using the accession numbers listed in S1 Table. All generated structural models can be downloaded

from S2 Data. All other relevant data are within the paper and its Supporting Information files.

**Funding:** This work was supported by l'Agence Nationale de la Recherche grant ANR-21-CE11-0001-01 to M.K. A.B. was supported by a postdoctoral fellowship from Fondation Recherche Médicale (FRM). E.V.K. is supported by funds of the National Institutes of Health of USA (National Library of Medicine) Intramural Research Program. V.V.D. was partially supported by the National Institutes of Health of USA (National Library of Medicine) Visiting Scientist Fellowship. The funders had no role in study design, data collection and analysis, decision to publish, or preparation of the manuscript.

**Competing interests:** The authors have declared that no competing interests exist.

**Abbreviations:** AU, approximately unbiased; CP, capsid protein; HGT, horizontal gene transfer; HMM, hidden Markov model; HVT, horizontal virus transfer; lDDT, local distance difference test; MP, movement protein; PDB, Protein Data Bank; PLRV, potato leaf roll virus; RMSD, root-mean-square deviation; SJR, single jelly-roll; SPMV, satellite panicum mosaic virus; STNV, satellite tobacco necrosis virus; TMV, tobacco mosaic virus.

viruses spanning the boundaries of the virus realms and infecting widely diverse hosts including prokaryotes and eukaryotes. In a sharp contrast, proteins involved in virus–host interactions are typically host-specific and hence are restricted to relatively narrow groups of viruses, in most cases, within a virus family or order. Comparatively recent acquisition from the host is demonstrable for many of these proteins. A common route of evolution in this class of virus proteins is exaptation whereby a host or a virus protein is repurposed for a new function in virus–host interaction [8–10].

However, exceptions are known when homologous proteins mediate the interactions of diverse viruses with a particular group of hosts. A quintessential example are the movement proteins (MPs) of plant viruses that help the viruses move through plasmodesmata, membranous channels in plant cell walls [11,12]. The plasmodesmata are permeable for small molecules but have a size exclusion limit that precludes free passage of larger molecules, such as most proteins and RNA, and macromolecular complexes, such as virus particles [13]. Although the properties of the plasmodesmata vary widely across different plant cell types and species, typically, active transport mechanisms are required for the passage of large molecules and particles. Therefore, most plant viruses, to the exclusion of capsid-less *Endornaviridae*, *Narnaviridae*, and *Mitoviridae* which are vertically transmissible RNA replicons [14,15], encompass genes or gene blocks encoding dedicated MPs that mediate virus passage across plasmodesmata. The MPs have been shown to bind the virus genome RNA or DNA and increase the size exclusion limit of the plasmodesmata, providing channels for passage of virions or virus genomes [16,17].

The most common MPs by far belong to the so called 30K superfamily that spans 2 realms of viruses, *Riboviria* including its both kingdoms, *Orthornavirae* and *Pararnavirae*, and *Monodnaviria*. More specifically, the 30K superfamily MPs are encoded by numerous families of RNA viruses within *Orthornavirae* (*Alphaflexiviridae*, *Secoviridae*, *Betaflexiviridae*, *Tombusviridae*, *Bromoviridae*, *Virgaviridae*, *Tospoviridae*, *Botourmiaviridae*, *Fimoviridae*, *Phenuiviridae*, *Aspiviridae*, *Kitaviridae*, *Mayoviridae*, and *Rhabdoviridae*) and the expansive families *Caulimoviridae* within *Pararnavirae* and *Geminiviridae* within *Monodnaviria* [18,19]. Although viruses encoding 30K MPs were previously only described in "higher" vascular plants, 30K MP-like sequences closely related to those of nepoviruses [20] and ophioviruses [21] were recently found in moss (*Bryopsida*, *Selaginellaceae*, *Lycopodiaceae* [nonvascular plants]), liverworts (*Lepidoziaceae* and *Anastrophyllaceae* [nonvascular plants]), and fern (*Vittaria lineata*, *Cyrtomium fortunei*, and *Lonchitis hirsute* [vascular plants]), basal plant lineages that were pivotal to the land plant evolution [22–24].

The prototype of the 30K superfamily is the MP of tobacco mosaic virus (TMV), a positive-sense RNA virus of the *Virgaviridae* family, the classical experimental model of virology (the name of the 30K superfamily comes from the molecular mass of the TMV MP, 30 kDa) [18,25]. The TMV MP is an RNA-binding protein that forms a ribonucleoprotein with the virus genomic RNA that is transported through the plasmodesmata by increasing the size exclusion limit [26–28], a mechanism likely used by most members of the 30K superfamily. However, some members of the 30K MP superfamily have been shown to change the size exclusion limit of the plasmodesmata via a different mechanism, namely, by forming tubular structures that mediate virion trafficking [19,29–31] or through interaction with virus capsid proteins (CPs) [29,32].

The broad conservation of the 30K MPs across diverse families of plant viruses including unrelated ones belonging to different realms implies the spread of the MP genes via horizontal gene transfer (HGT). However, the ultimate origin of the 30K MPs remains enigmatic because no homologs of these proteins have been detected by sequence similarity searches, even using the most sensitive of the available methods, whereas tertiary structures of the MPs have not

been determined. Comparison of the predicted secondary structure elements suggested that the 30K MPs share a common core that consists of 7 to 8 β-strands [18,19,25], and it has been noted that this core domain might be related to the single jelly-roll (SJR) fold found in CPs of numerous small viruses with icosahedral capsids [19]. However, because these predictions were not statistically significant, it remained unclear whether the similarities between CPs and MPs reflected homology [19].

Recently, protein structure prediction has been revolutionized by high performance machine learning-based methods, AlphaFold2 (AF2) and RoseTTAFold [33,34]. These methods consistently yield accurate structure predictions for globular proteins with many diverse homologs. We took advantage of these tools to probe the origin of the 30K superfamily of MPs. Comparisons of the AF2 models of the MPs to the available protein structures unequivocally demonstrated close structural similarities between the MPs and virus jelly-roll CPs. We therefore conclude that the 30K MPs evolved via ancient duplication of the SJR CP gene followed by exaptation for the movement function.

## Methods

### Representative protein sequences of 30K MPs, clustering and phylogenetic analysis

Sequences of 18 representative 30K MP superfamily proteins [19] (S1 Table) were downloaded from GenBank and used as queries in sequence similarity searches performed with blastp [35] against the nr_vir70_1_Nov database (E-value cutoff of 0.001) [36]. The retrieved sequences of MP homologs were clustered to 90% minimum sequence identity using MMseqs2 [37]. The resulting dataset was used for clustering analysis using CLANS [38] and maximum-likelihood phylogenetic analysis with IQ-TREE 1.6.12 using the options -m TEST -bb 1000 -alrt 1000 [39]. CLANS analysis, where the sequences are positioned in a multidimensional space based on the strength of their pairwise similarities, was performed with the PSI-BLAST option and e-value of $10^{-3}$. The clusters were identified at *P-value* = $10^{-15}$. For the maximum-likelihood analysis and the identification of the D motif in SJR CPs, the sequences were aligned with PROMALS3D [40] with default parameters. In the alignment used for the maximum-likelihood analysis, poorly aligned (low information content) positions were removed using trimal with the -gt 0.2 option [41]. Phylogenetic analysis was performed using IQ-TREE [39], with the protein substitution model detection. The tree was rooted at midpoint and visualized in iTOL [42]. Sensitive profile–profile comparisons for remote sequence similarity detection were performed using HHsearch [43] against the Protein Data Bank (PDB) database.

The local charge distribution plots of the selected MPs and SJR CPs were obtained with the help of the "*chargeCalculationLocal*" option in the "*idpr*" package in R, using window size option 21 [44].

### Three-dimensional structure prediction and analysis

The 18 MP sequences selected to represent different virus families and the 8 MP sequences from viruses associated with mosses, liverworts, and ferns [20,21] were used as inputs for AF2 (version 2.1.1, [33]) and RoseTTAFold [34] structure prediction. In particular, we used RoseTTAFold when MP structures modeled by AF2 were of poor quality, as estimated using the local distance difference test (lDDT) [45]. The quality of the RoseTTAFold models was assessed using residue-wise CA-lDDT implemented in the end-to-end version of RoseTTAFold.

Structure-based searches were performed with the DALI [46] server, and structural similarities between the MPs and their homologs were evaluated based on the DALI Z scores. The Z

score measures the quality of the structural alignment, with scores above 2 generally considered significant. The structural matches were further evaluated by superimposition of the structures using the MatchMaker algorithm implemented in University of California, San Francisco (UCSF) Chimera [47], followed by visual inspection. The top 20 hits against the PDB50 database in DALI searches were extracted and used for all-against-all structure comparisons on the DALI server. As an additional structural alignment tool, we used MUSTANG [48] to generate a pairwise similarity matrix based on root-mean-square deviation (RMSD) values between all modeled MP structures and related CPs. Similarity matrices that were generated from the DALI and MUSTANG comparisons were used in "pvclust" R package version 2.2.-0 [49] to generate a dendrogram with bootstrap supports (approximately unbiased (AU) *p*-value was computed by multiscale bootstrap resampling) from a similarity matrix by average linkage clustering. The heatmaps were plotted using the "pheatmap" R package version 1.0.12 [50]. Different clustering methods were tested (average, complete, ward.D, single, mcquitty linkage methods) and the cophenetic correlation coefficient was calculated for all to determine the clustering method that best represented the data. The complete linkage clustering method proved to be the best choice with respect to the correlation coefficient values and biological interpretation of the clusters.

## Results and discussion

### Horizontal spread of the 30K superfamily movement proteins across the plant virome

The representative MPs (*n* = 18; S1 Table) that belong to the 30K superfamily were used as queries in one iteration of protein BLAST search against the virus database filtered to 70% identity (nr_vir70). The MP sequences detected during the parallel searches were dereplicated and clustered at 90% sequence identity, yielding 389 clusters of related sequences, representing 16 virus families. Representatives from each cluster were then subjected to CLANS analysis (S2 Table) to identify more coarse-grained clusters (Fig 1A). CLANS detected 16 clusters of MPs that mostly corresponded to virus families and grouped into 5 superclusters, including: (1) *Geminiviridae* (realm *Monodnaviria*); (2) *Aspiviridae*, *Fimoviridae*, and *Phenuiviridae* (phylum *Negarnaviricota* within *Orthornavirae*); (3) *Kitaviridae*, *Bromoviridae*, *Mayoviridae*, *Alphaflexiviridae*, *Virgaviridae (Furovirus)*, *Tombusviridae (Umbravirus)* (phylum *Kitrinoviricota* within *Orthornavirae*), and *Tospoviridae* (*Negarnaviricota*); (4) *Rhabdoviridae* (*Negarnaviricota*), *Caulimoviridae* (*Pararnavirae*), *Virgaviridae (Tobamovirus)*, *Betaflexiviridae*, and *Secoviridae*; (5) *Tombusviridae* (*Tombusvirus* and *Aureusvirus*) (*Kitrinoviricota*), and *Botourmiaviridae* (*Lenarviricota* within *Orthornavirae*) families (Fig 1A). Superclusters 1, 2 and 5 were homogeneous, each including 1 or several related virus families, but superclusters 3 and 4 each included highly diverse, distantly related viruses, implying multiple HGT events. Notably, different genera of the families *Virgaviridae* and *Tombusviridae* did not cluster together, unlike other virus families, and were represented in both superclusters 3, 4 and 5 suggestive of relatively recent HGT and non-orthologous (although homologous) MP gene replacements. Furthermore, among the virgaviruses, some encode a 30K MP, whereas others encompass the so-called triple gene block MPs [51], demonstrating exchangeability of unrelated movement machineries.

To further explore the relationships among the 30K MPs, the 389 MP sequences representing the 90% identity clusters were aligned using PROMALS3D [40], and a maximum-likelihood phylogenetic tree was constructed (Fig 1B and S1 Data). Overall, the tree topology recapitulated the results of CLANS analysis, with clusters and superclusters forming clades, mostly, with high bootstrap support (Fig 1B).

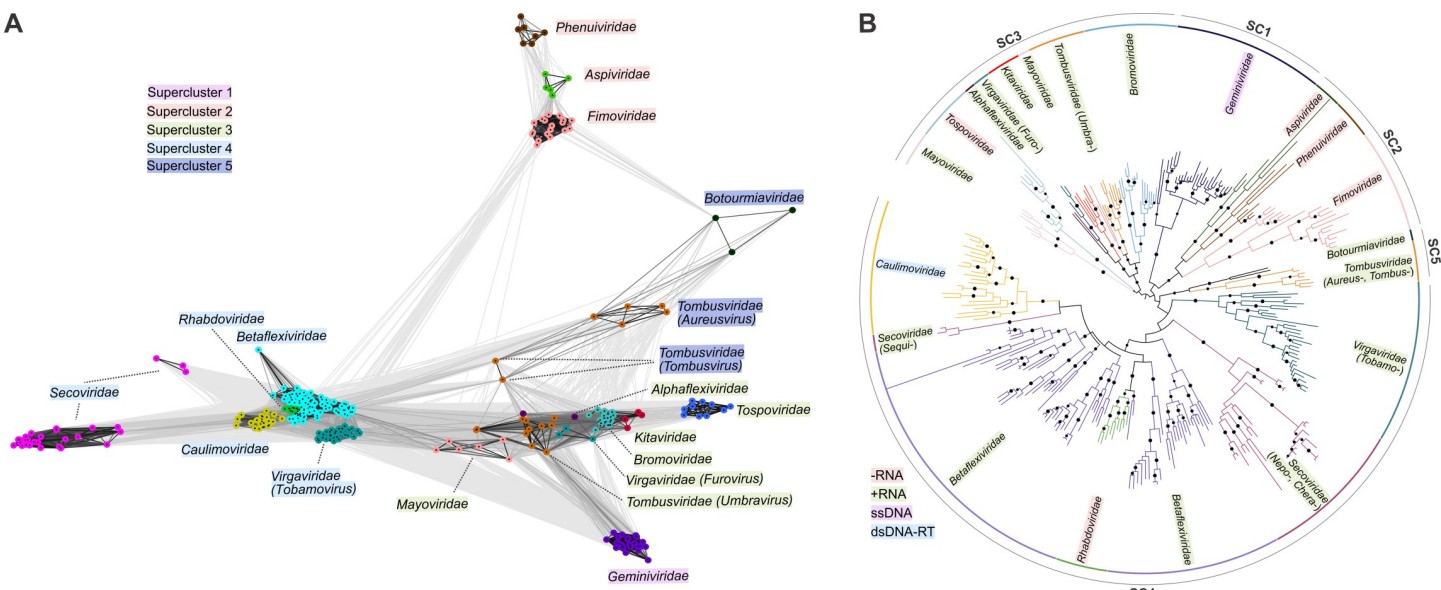

**Fig 1. Sequence similarity and phylogeny of the 30K MPs.** (**A**) Clustering of 30K MP sequences by pairwise sequence similarity (CLANS *P*-value $\leq 1 \times 10^{-15}$). The clusters are colored and named by virus families, while the outline boxes indicate if the virus family is part of superclusters 1–4. The lines represent sequence relationships, darker colors indicate closer sequence similarity. The HSP values used for clustering can be found in S2 Table. (**B**) Maximum-likelihood phylogenetic tree of 30K MP sequences obtained by IQ-TREE. SC, supercluster. The circles at the nodes indicate bootstrap branch support values ≥90. Superclusters 1–5 are also indicated. The tree in newick format can be found in S1 Data. HSP, high scoring pair; MP, movement protein.

Phylogenetic analysis further confirmed multiple horizontal exchanges of the MP genes during the evolution of plant viruses. For example, reverse-transcribing caulimoviruses, negative-sense (-)RNA viruses of the family *Rhabdoviridae* (genus *Cytorhabdovirus*) and positive-sense (+)RNA viruses of the family *Secoviridae* (genus *Sequivirus*) are nested among MPs of *Betaflexiviridae*, another family of (+)RNA viruses, suggesting that the latter virus group acted as a superspreader of the MP genes during the evolution of plant viruses. By contrast, (-)RNA *Tospoviridae* cluster with different families of (+)RNA viruses in the supercluster 3, with the rest of (-)RNA viruses forming a disconnected clade corresponding to supercluster 1 (Fig 1B), suggesting at least 3 independent MP introduction events into (-)RNA viruses. An extensive shuffling of the MP genes is also observed in the families *Tombusviridae* and *Virgaviridae*, where viruses from different genera form paraphyletic groups in the phylogeny.

## The 30K MPs are homologous to the single jelly-roll capsid proteins

No high-resolution structure is available for any member of the 30K MP superfamily. Thus, to gain insights into the deeper evolutionary history of the 30K MPs through structure-based homology searches, we leveraged the state-of-the-art high performance structural modeling methods AF2 and RoseTTAFold [33,34]. The quality of the obtained 30K MP structural models, assessed using the lDDT [45], was found to be generally high in the conserved central region of the proteins, whereas the variable terminal regions were often unstructured and therefore modeled with a lower quality (S1 Fig and S3 Table and S2 Data). The well-structured central region was found to adopt the jelly-roll fold (Fig 2A) consisting of 8 antiparallel β-strands, typically denoted B through I, that form 2 juxtaposed β-sheets, composed of BIDG and CHEF strands, respectively (Fig 2A and 2B) [52,53]. The jelly-roll domain was readily identifiable in MPs encoded by viruses from all analyzed virus families (Fig 2B), with the molecular mass of the core jelly-roll domain varying between 14.6 kDa for tomato spotted wilt

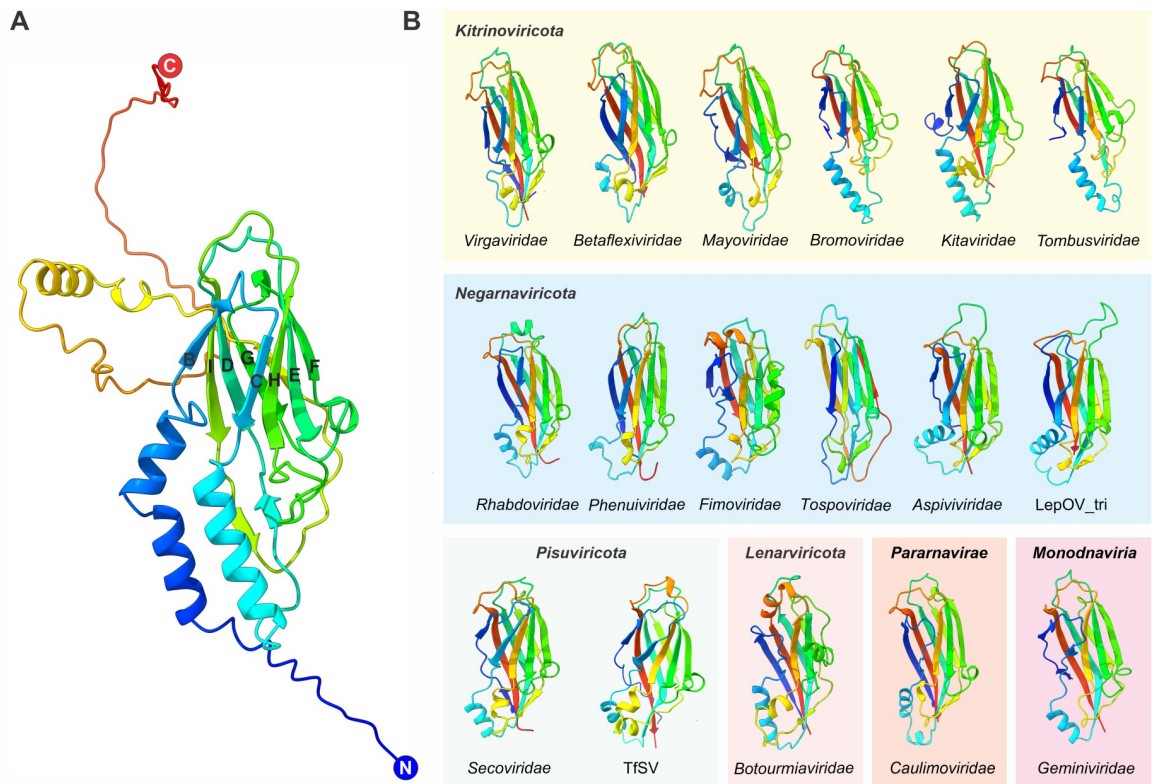

**Fig 2. Structural modeling of 30K MPs.** (**A**) Structural model of a representative full-length MP of the cabbage leaf curl virus (family *Geminiviridae*). The structure is colored using the rainbow scheme from blue (N-terminus) to red (C-terminus). The β-strands of the jelly-roll domain are indicated with Roman letters. (**B**) Structural models of the 30K MPs representing different virus families. The variable terminal ends were trimmed for the convenience of presentation. The structures are colored using the rainbow scheme from blue (N-terminus) to red (C-terminus). The structures are grouped according to established virus taxonomy. In the case of *Orthornavirae*, the corresponding phyla are indicated. Phylum *Kitrinoviricota*: *Virgaviridae* is represented by TMV, *Betaflexiviridae* by actinidia virus, *Mayoviridae* by raspberry bushy dwarf virus, *Bromoviridae* by cucumber mosaic virus, *Kitaviridae* by citrus leprosis virus C, *Tombusviridae* by carrot mottle virus; phylum *Negarnaviricota*: family *Rhabdoviridae* is represented by lettuce necrotic yellows virus, *Phenuiviridae* by rice stripe virus, *Fimoviridae* by rose rosette virus, *Tospoviridae* by tomato spotted wilt virus, *Aspiviridae* is represented by citrus psorosis virus and lepidozia ophiovirus tri (LepOV_tri) associated with hairy liverwort; phylum *Pisuviricota*: *Secoviridae* is represented by cherry rasp leaf virus and tomato fern seco-like virus (TfSV); phylum *Lenarviricota*: family Botourmiaviridae is represented by ourmia melon virus. Family *Caulimoviridae* (kingdom *Pararnaviae*) is represented by cauliflower mosaic virus, whereas family *Geminiviridae* (realm *Monodnaviria*) is represented by cabbage leaf curl virus. The PDB structure files for the modeled MPs can be found in S2 Data. MP, movement protein; PDB, Protein Data Bank; TMV, tobacco mosaic virus; TfSV, tomato fern seco-like virus.

virus (*Tospoviridae*) and 19 kDa for parsnip yellow fleck virus (*Secoviridae*), accounting for about half of the entire mass of the corresponding MPs.

The structural models of representative MPs were used as queries in DALI searches of the PDB database of protein structures. These searches retrieved as best hits the SJR CPs from diverse icosahedral viruses of eukaryotes, with significant Z scores ranging from 6.2 to 9.9 (S1 Table). The majority of the best hits were to CPs of the family *Tombusviridae* (S1 Table). However, the MPs of the viruses in the families *Caulimoviridae*, *Betaflexiviridae* (*Vitivirus*) and *Virgaviridae* produced the same highest scoring hit to the CP of satellite panicum mosaic virus (SPMV, *Papanivirus*; S1 Table). The rest of the hits were to CPs of viruses from other families, largely associated with plant hosts, but also including some animal viruses, such as those of the families *Astroviridae* and *Hepeviridae* (S1 Table).

Structural comparison of the 30K MPs and SJR CPs revealed closely similar jelly-roll topologies (Figs 3A and S2). The α-helix between G and H β-strands (not part of the canonical jelly-roll fold)

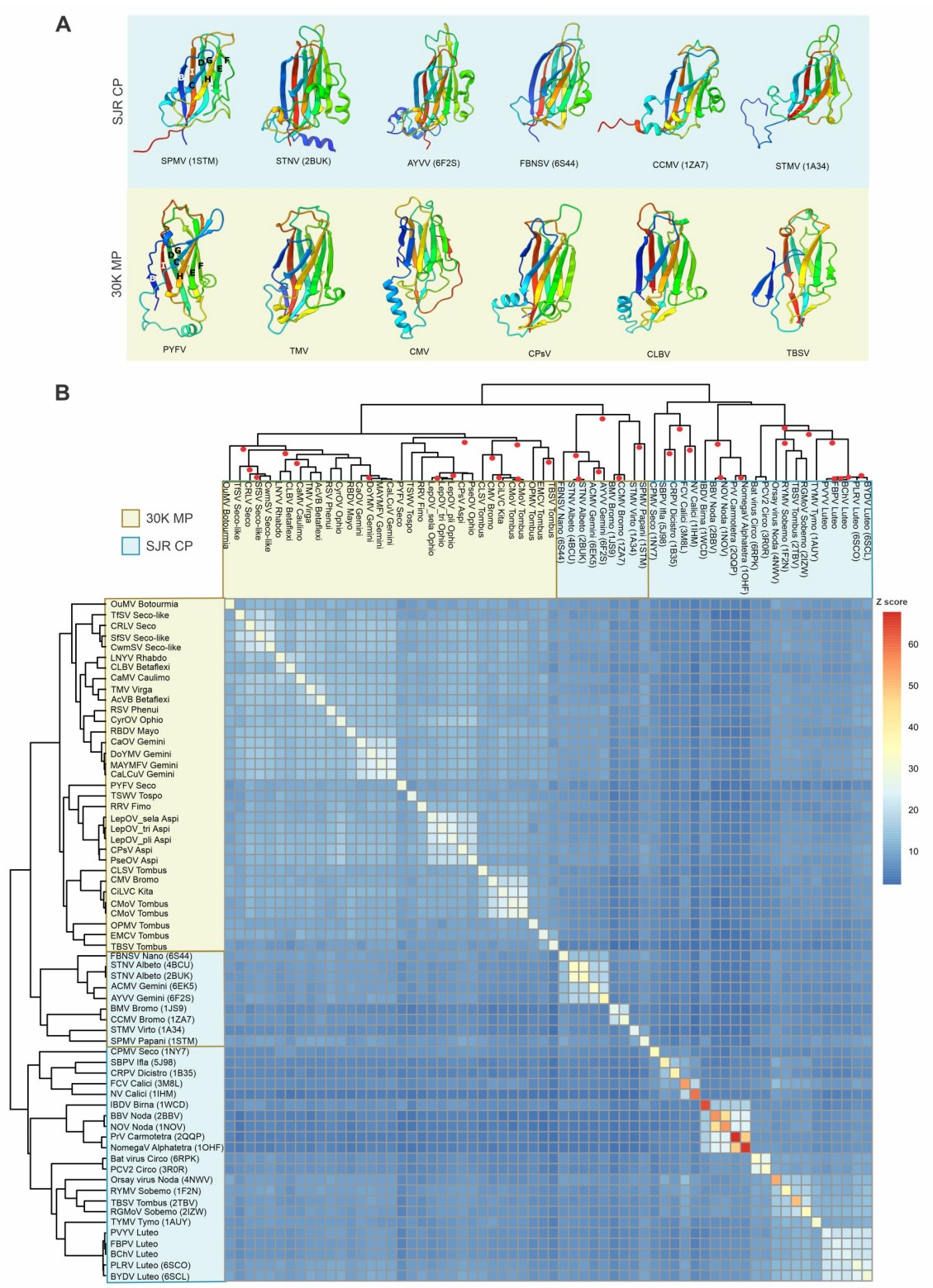

**Fig 3. Structural similarity between SJR CPs and 30K MPs.** (**A**) Structures of the SJR CPs homologous to 30K MPs obtained after a DALI search of PDB database, in the upper row highlighted with a blue background. The bottom row shows the jelly-roll region for the selected structures of 30K MP representatives, highlighted with a yellow background. The first structures on the utmost left in the upper and bottom row have the BIDG-CHEF β-strands annotated. The structures are colored using the rainbow scheme from blue (N-

terminus) to red (C-terminus). (**B**) Dendrogram and heatmap of complete linkage clustering of 30K representatives and SJR CPs. The red circles indicated in the top dendrogram, represent bootstrap values ≥90 obtained with R package "pvclust." The CPs and MPs are indicated in blue and yellow, respectively. Structures of 30K MPs and SJR CPs belong to: BMV, CCMV, FBNSV, STNV, ACMV, AYVV, STMV, SPMV, IPNV, IBDV, BBV, NoV, PrV, NomegaV, BFDV, PCV2, PhMV, TYMV, BYDV, BChV, PVYV, FBPV, RGMoV, RYMV, SBMV, TNV, BPMV, CPMV, FCV, NV, HRV16, SBPV, and CrPV. The newick format of the dendrogram obtained in DALI can be found in S3 Data. ACMV, African cassava mosaic virus; AYVV, ageratum yellow vein virus; BBV, black beetle virus; BChV, beet chlorosis virus; BFDV, beak and feather disease virus; BMV, brome mosaic virus; BPMV, bean pod mottle virus; BYDV, barley yellow dwarf virus; CCMV, cowpea chlorotic mottle virus; CP, capsid protein; CPMV, cowpea mosaic virus; CrPV, cricket paralysis virus; FBNSV, faba bean necrotic stunt virus; FBPV, faba bean polerovirus 1; FCV, feline calicivirus; HRV16, human rhinovirus; IBDV, infectious bursal disease virus; IPNV, infectious pancreatic necrosis virus; MP, movement protein; NomegaV, nudaurelia capensis omega virus; NoV, nodamura virus; NV, Norwalk virus; PCV2, porcine circovirus 2; PDB, Protein Data Bank; PhMV, physalis mottle virus; PrV, providence virus; PVYV, pepper vein yellows virus; RGMoV, ryegrass mottle virus; RYMV, rice yellow mottle virus; SBMV, southern bean mosaic virus; SBPV, slow bee paralysis virus; SJR, single jelly-roll; SPMV, satellite panicum mosaic virus; STMV, satellite tobacco mosaic virus; STNV, satellite tobacco necrosis virus; TNV, tobacco necrosis virus; TYMV, turnip yellow mosaic virus.

found in many MPs is also present in the SJR CPs of bromoviruses and solemoviruses as well as geminiviruses (e.g., ageratum yellow vein virus, PDB: 6F2S) and satellite tobacco necrosis virus (STNV, *Albetovirus*, PDB: 4BCU), suggesting a closer evolutionary relationship between the MPs and the CPs of these plant viruses. The consistent, significant structural similarity between the MPs and SJR CPs, and in particular, the same topology of the jelly-roll domains indicate that the 2 groups of proteins are indeed homologous. The SJR CPs are ubiquitous among the numerous groups of riboviruses and monodnaviruses with icosahedral capsids that infect diverse unicellular and multicellular eukaryotes from at least 9 eukaryotic kingdoms [52,54]. By contrast, the 30K MPs show a broad but scattered spread among viruses that primarily infect plants, i.e., restricted to a single eukaryotic kingdom, Chloroplastida, or in some cases, plants and their vectoring organisms. Thus, it appears highly likely that the 30K MPs evolved from the CPs.

To further analyze the evolutionary relationships between 30K MPs and SJR CPs, we performed an all-against-all comparison of the 30K MP structural models and SJR CP structures identified through the DALI searches. To avoid potential artifacts caused by the variable terminal regions of the MPs that have no counterparts in the CPs, for this analysis, only the jelly-roll domains of the MPs were considered. In the dendrogram obtained from the DALI Z scores, all MPs formed a single clade that was lodged within the diversity of the CPs (Fig 3B and S3 Data), suggesting monophyly of the 30K MP superfamily. The MP clade clustered with a distinct CP subclade that includes plant satellite RNA viruses, *Geminiviridae*, *Nanoviridae*, and *Bromoviridae* (Fig 3B). All these viruses infect plants and have highly compact SJR CP structures [55–57]. Given that satellite viruses are relatively rare in plant infections, the CPs of *Geminiviridae*, *Nanoviridae*, and *Bromoviridae* families seem to be the more likely ancestors of the 30K MPs. Furthermore, the CPs of bromoviruses and geminiviruses share with the 30K MPs the characteristic α-helical insertions within the jelly-roll domain. To corroborate these results, we used MUSTANG, an algorithm that aligns residues on the basis of similarity in patterns of both residue–residue contacts and local structural topology, creating a multiple structural alignment [48]. The dendrogram resulting from hierarchical clustering of the structural similarity values obtained with MUSTANG was largely congruent with that produced by DALI (S3 Fig and S4 Data). In this dendrogram, the MPs were nested within the CP diversity and formed a sister group to the CPs of the same assemblage of plant RNA and DNA viruses (geminiviruses, nanoviruses, bromoviruses, satellite viruses) as in the Z-score-based dendrogram, with the only notable difference being that the CP of SPMV was placed among the MPs. The latter placement is likely due to poor representation of the SPMV CP group (only 1 structure with no homologs identifiable at the sequence level) as well as a genuine high structural similarity to the MPs (S1 Table). Although we consider it unlikely that the SPMV CP evolved from an MP, this possibility cannot be formally ruled out.

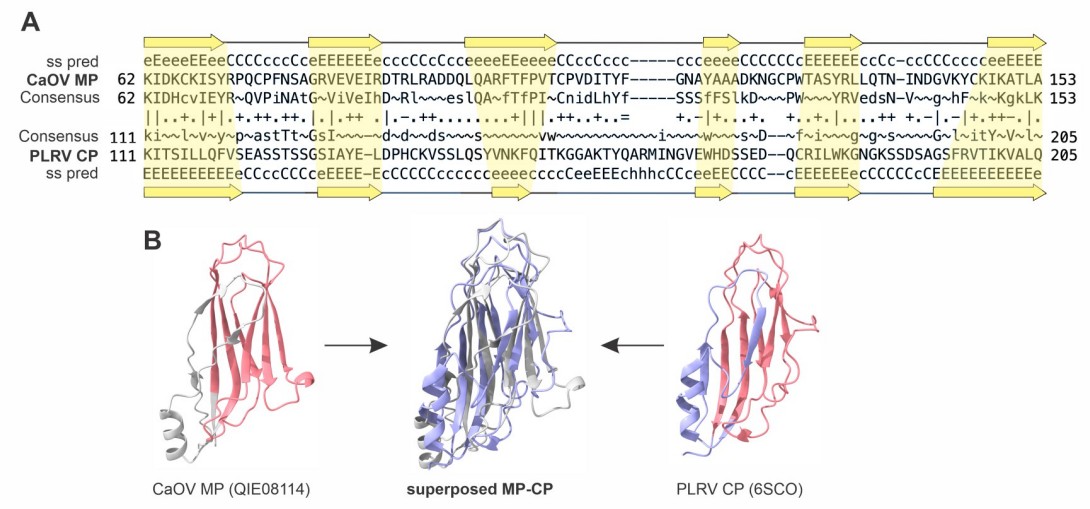

**Fig 4. Validation of the homology between SJR CPs and 30K MPs by sensitive sequence analysis.** (**A**) Homologous regions between the CP of PLRV (PDB ID: 6SCO) and Camellia oleifera geminivirus (CaOV) 30K MP (accession number: QIE08114) obtained with HHsearch analysis against the PDB database. Secondary structure prediction is indicated by arrows for beta strands in yellow. (**B**) The structural model of CaOV 30K MP and PLRV CP. The homology region between the 2 proteins found in HHsearch against the PDB database is shown in red. The superposition of the conserved jelly-roll regions of CaOV 30K MP and PLRV CP is shown in the middle. The PLRV CP is colored light purple, and the CaOV 30K MP is colored light gray. CP, capsid protein; MP, movement protein; PDB, Protein Data Bank; PLRV, potato leaf roll virus; SJR, single jelly-roll.

Initial sequence similarity searches using BLASTP queried with 30K MP sequences yielded no significant matches outside the 30K superfamily, consistent with previous analyses [19]. However, in retrospect, after discovering the structural similarities between the MPs and SJR CPs, we reexamined this relationship using more sensitive comparisons of profile hidden Markov models (HMMs). Searches queried with the profile HMMs of 30K MPs against the profile HMMs of the PDB database yielded matches between the MPs of geminiviruses and SJR CP of potato leaf roll virus (PLRV, *Solemoviridae*; PDB ID: 6SCO), with significant probability scores (>90%, Fig 4A). The aligned regions mapped within the jelly-roll domains of the 2 proteins (Fig 4A). Consistently, the corresponding regions of the PLRV CP and geminivirus MP structural models, including the α-helix between β-strands C and D, could be superposed (Fig 4B). Thus, geminivirus MPs appear to more closely resemble the ancestral state of the 30K MP superfamily, with the relationship between the MPs and CPs still detectable at the sequence level. Phylogenetic and clustering analyses suggest that, following the divergence from the ancestral SJR CP, geminivirus MPs largely evolved vertically, without interfamilial horizontal exchange with other plant virus families (Fig 1), which conceivably contributed to the conservation of the ancestral features. We note, however, that the potentially archaic features of the geminivirus MPs do not necessarily imply that these proteins are ancestral to the 30K MPs of other viruses. Indeed, the vast virome of the vascular, particularly flowering plants, is dominated by RNA viruses of the kingdom *Orthornavirae* [58], suggestive of their rapid co-diversification and long coevolution with their hosts. Thus, a scenario under which the ancestral 30K MP gene was hosted by RNA viruses appears more parsimonious.

## The MPs of multicellular algae and nonvascular plants

Ultimately, identification of the origins of the 30K MPs requires understanding the coevolution of contemporary plant virome with its plant hosts. It is generally recognized that

emergence and diversification of the plant virome occurred during terrestrialization of plants that apparently started with subaerial *Zygnematophyceae* freshwater algae followed by nonvascular terrestrial mosses and vascular plants [58,59]. The closest relatives of *Zygnematophyceae* algae for which viruses are known are algae of the genus *Chara*. The Charavirus canadiensis (CV-Can) and Charavirus australis (CV-Aus) viruses are 2 closely related, presumably rod-shaped (+)RNA viruses that encode TMV-like CPs along with the genes of unknown function that occupy the same genomic location as the 30K MP gene of TMV [60,61]. However, the proteins encoded by these genes exhibit no sequence similarity to any of the known MPs or other proteins. Notably, our AF2 modeling showed that the core structure of these *Chara* virus proteins was closely similar to that of the CPs of flexible filamentous viruses, such as alphaflexiviruses (S4 Fig). It seems likely that these proteins of *Chara* viruses evolved from capsid proteins of filamentous viruses to facilitate virus movement between *Chara* cells through the distinct algal plasmodesmata unrelated to those of vascular plants [11]. This evolutionary scenario parallels the exaptation of SJR CPs for the movement function of the 30K MPs, an analogy further strengthened by the acquisition of N-terminal extension observed in both cases (see below). Whether this protein functions in virus movement in *Chara* algae, remains to be validated experimentally.

As established previously, viruses encoding 30K MPs are present in lower vascular plants (ferns and lycophytes) and are ubiquitous in gymnosperms and angiosperms [19,62]. Recently, such 30K MP-encoding viruses were not only confirmed in ferns, but also found in nonvascular plants, namely, mosses and liverworts [20,21]. To address the possibility that the 30K MPs from moss, liverwort, and fern viruses resemble the ancestral state, we modeled their structures from secoviruses associated with common water moss (*Fontinalis antipyretica*), shoestring fern (*Vittaria lineata*), and tomato fern (*Lonchitis hirsuta*) [17] as well as from ophioviruses associated with hairy liverwort (*Lepidozia trichodes*) and basket liverwort (*Plicanthus hirtellus*), holly fern (*Cyrtomium fortunei*), Krauss' spike moss (*Selaginella kraussiana*), and Slender bog club-moss (*Pseudolycopodiella caroliniana*) [21]. All secovirus MPs grouped tightly with the MPs of the viruses of angiosperms in the genus *Nepovirus*, family *Secoviridae* (Fig 2B). Similarly, MPs of ophioviruses associated with nonvascular and lower plants clustered with the MP of angiosperm-infecting citrus psorosis ophiovirus (family *Aspiviridae*). Notably, besides the SJR domain, all ophiovirus MPs shared a characteristic C-terminal domain (PF11330; 30K_MP_C_Ter; HHpred probability = 98.3%) that is exclusive to ophiovirus 30K MPs. These observations, consistent with the previous phylogenetic analysis [20], suggest horizontal virus transfer (HVT) to lower vascular and nonvascular plants following the diversification of the *Secoviridae* and *Aspiviridae* families in angiosperms rather than emergence of 30K MPs in nonvascular mosses or liverworts that lack PDs.

## Possible functions of the core and terminal regions of the 30K MPs in virus movement

The N- and C-terminal regions of the 30K MPs are predicted to be largely disordered, without recognizable folded domains, and vary dramatically both within and between different virus families (Fig 5A and S4 Table), which can explain the lower quality of the structural models in these regions (S1 Fig). The length of the N-terminal extensions (relative to the jelly-roll domain) varies from 9 amino acid residues (aa) in geminiviruses to 130 aa in mayoviruses, whereas the C-terminal extensions vary from 12 to 289 aa, in betaflexiviruses alone. The N-terminal region has been implicated in tubule polymerization and plasmodesmatal targeting of the MP [25,63], whereas the C-terminal region appears to be predominantly responsible for the interactions with CPs, virions, virion packaging into tubules, and long-distance movement

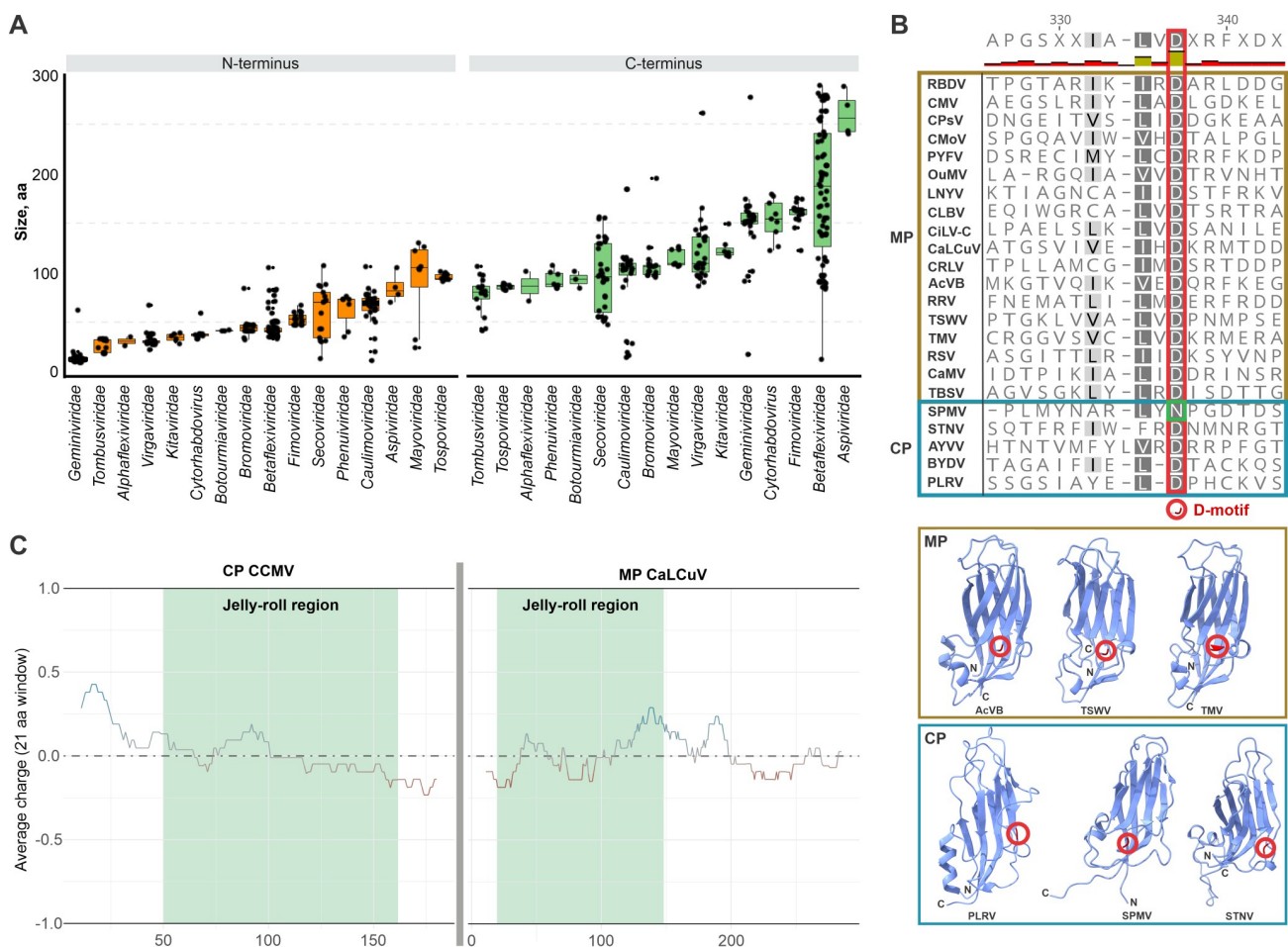

**Fig 5. Length variation of the terminal regions of the 30K MPs, D motif conservation and charge distribution in the 30K MPs and SJR CPs.** (A) Boxplot of the lengths of the N and C-terminal regions of 30K MPs. Orange boxes indicate values for N-terminal sizes and the green boxes indicate the C-terminal sizes. The x-axis denotes virus families and the y-axis the size of terminal ends by the number of amino acids. All the values are ordered by size from the smallest to the largest. The numeric values corresponding to the lengths of the N- and C-termini used for the boxplot can be found in S4 Table. (B) Top: the D motif region in the alignment of representative 30K MPs and SJR CPs. Note that in SPMV, the aspartate (D) is conservatively substituted with an asparagine (N). Bottom: the position of the D motif mapped on the MP and CP protein structures. The D motif is marked with a red circle. (C) Local charge distribution for CaLCuV 30K MP and CCMV SJR CP (PDB: 1ZA7) sequence by amino acid residue position (window size 21). The jelly-roll region is represented by a light green box. The height of the line above the gray threshold (0.0) indicates the value of the positive charges. The numerical values used to plot the charge distributions can be found in S5 Table. CCMV, cowpea chlorotic mottle virus; CP, capsid protein; MP, movement protein; SJR, single jelly-roll; SPMV, satellite panicum mosaic virus.

[32,64–69]. Overall, viruses in the families *Aspiviridae*, *Betaflexiviridae*, *Fimoviridae*, *Rhabdoviridae*, and *Geminiviridae* have longer N-terminal regions compared to the rest of the MPs, but this does not seem to correlate with the tubule formation (Fig 5A). The C-terminal MP regions are equally variable in size (Fig 5A), but again, there is no obvious correlation between the size of the extensions and the reported interactions between the C-termini of MPs with the respective virus CPs.

The most conserved feature of the 30K MPs is the D-motif [19] that includes a conserved aspartate residue located between β-strands E and F (marked dark red in Fig 5B), consistent with previous predictions on the position of the D-motif between 2 β-strands [18,19,25]. Alignment of the representative 30K MPs and SJR CPs reveals a degree of conservation of the D-motif in SJR CPs, particularly in CPs with the closest structural similarity to the MPs,

including geminiviruses, bromoviruses, and some satellite viruses (Figs 5B and S5). The sporadic presence of the D-motif in SJR CPs is consistent with a scenario under which MPs evolved from a specific group of CPs that contain this motif, rather than from a more ancient common ancestor with CPs.

Positively charged N-terminal regions of SJR CPs, commonly known as R-arms, bind viral RNA, or DNA genomes, promoting virion formation [70–72]. Similarly, positive charges have been shown to be required for nucleic acid binding by the 30K MPs [70,73–75]. However, whereas in SJR CPs, the positive charges involved in nucleic acid binding concentrate in the unstructured R-arms preceding the jelly-roll domain, in the 30K MPs, positively charged patches are distributed across the jelly-roll domain itself or the C-terminal extensions with no counterparts in the CPs (Figs 5C and S6 and S5 Table). We hypothesize that positive charge redistribution played an important role in the evolution of the CP into MP, facilitating the formation of distinct virus genome-MP complexes capable of passing through the plasmodesmata.

## Evolution of plant virus movement proteins

Our results suggest that the 30K MPs originated from a distinct group of the SJR CPs (Fig 6). The viruses that encoded the ancestral SJR CP at the origin of the 30K MPs might no longer be part of the contemporary virome. Thus, it might not be possible to pinpoint with confidence the actual ancestor. Regardless of the exact identity of the ancestral virus, we hypothesize that the 30K MPs emerged in a virus that infected multicellular freshwater algae during their evolution on the route to nonvascular and later vascular land plants. After a chance duplication of the original SJR CP gene, exaptation of one of the copies for the movement function provided a strong fitness advantage by facilitating efficient spread of the virus through evolving plasmodesmata (Fig 6). The following rapid horizontal spread of the MP gene among emerging plant viruses with different genome types drove the diversification of the 30K MP superfamily and the dramatic expansion of the global plant virome.

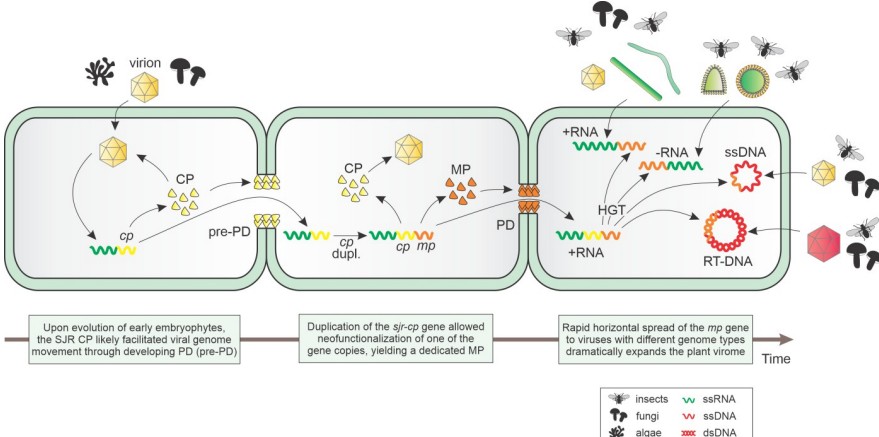

**Fig 6. An evolutionary scenario for the origin of the 30K MP superfamily from SJR CPs.** The ancestral virus is predicted to have an RNA genome (green wavy line) and encode an SJR CP, which was responsible for capsid formation and promoted intercellular movement through developing plasmodesmata. Duplication and neofunctionalization of the *cp* gene (yellow wavy line) led to the emergence of a dedicated *mp* gene (orange wavy line). Subsequently, the *mp* gene was horizontally transferred to other RNA viruses and viruses with DNA genomes (red wavy lines). Abbreviations: CP, capsid protein; (pre-)PD, (developing) plasmodesmata; MP, 30K movement protein; dupl., gene duplication; SJR, single jelly-roll.

The diversity of the contemporary plant virome that is dominated by RNA viruses remains to be a subset of the invertebrate RNA virome diversity [76,77]. Therefore, it appears most likely that the invertebrate virome seeded the plant virome through HVT enabled by plant-feeding nematodes and arthropods that currently serve as vectors for plant viruses. The expansion of the plant virome was contingent on the acquisition of MP, putting the horizontal spread of 30K MP among diverse virus families into the same timeframe. In support of this perspective, it was shown that the transgenic expression of the TMV MP in *Nicotiana benthamiana* enabled cell-to-cell and systemic movement of flock house virus, a single-stranded RNA insect virus not known to otherwise infect plants [78], providing experimental illustration of the critical role of MPs in the adaptation of insect viruses to plant hosts. Notably, the horizontal spread of the 30K MP gene placed it into widely different genome contexts including (+)RNA and (-)RNA viruses, reverse-transcribing viruses and single-strand DNA viruses. Furthermore, 30K MPs were combined with diverse virion architectures formed by the SJR CPs and several other, unrelated CPs as in the classic case of rod-shaped TMV or enveloped (-)RNA viruses. Conceivably, this diversity of the genomic contexts drove the functional and evolutionary diversification of the 30K MPs that remains to be explored in detail.

The route of 30K MP evolution represents a remarkable case of "intramural" exaptation, whereby a preexisting virus protein dramatically changed its function, providing strong selective advantage to the virus [10]. Notably, 3 divergent copies of non-jelly-roll CP of filamentous closteroviruses were exapted along a parallel route for distinct functions in virus capsid formation and transport [79]. One of the components of the triple gene block movement machinery, which represents an alternative to the 30K MPs [80], is a specialized superfamily 1 helicase, providing an additional example of functional exaptation of a preexisting virus protein for the function in virus movement. The exaptation of both the 30K MP and the helicase for enabling virus movement apparently involved addition of an extended unstructured N-terminal region that is important for the formation and transport of the virus nucleoprotein [51,81]. Finally, the putative MP of Chara viruses with an extended N-terminal domain and a core alphaflexivirus-like CP domain (S3 Fig) might represent yet another, independent case of CP exaptation for virus movement along a route similar to that of 30K MPs. These examples further emphasize exaptation as a key mechanism that shaped the virosphere ever since its inception and continues to contribute to virus diversification and evolution [6,10].

To conclude, this work demonstrates the potential of the new generation of protein structure prediction and analysis methods to illuminate key evolutionary events that remained out of reach of protein sequence-based analyses. Such findings, in turn, can be expected to inform further experimental studies.

## Supporting information

**S1 Table. Table of top single jelly-roll capsid protein hits in structural homology search with DALI using 30K movement proteins.**
(XLSX)

**S2 Table. High scoring pairs (HSP) values obtained by running psi-blast via CLANS and used for plotting the clustering network.**
(XLSX)

**S3 Table. plDDT values for all AlphaFold2 structural models.** Each excel sheet corresponds to a virus MP.
(XLSX)

**S4 Table. Sizes of N and C terminal MP ends per virus family used for the barplot.**
(XLSX)

**S5 Table. Distribution of local charge values for selected MPs in a window size 21, calculated with the "*chargeCalculationLocal*" option in the "*idpr*" package in R.**
(XLSX)

**S1 Data. The maximum-likelihood phylogenetic tree of the 30K MPs in newick format.**
(NWK)

**S2 Data. All MP AlphaFold models generated in this study.**
(ZIP)

**S3 Data. The dendrogram tree of 30K MPs and SJR CP hits obtained with DALI in newick format.**
(NWK)

**S4 Data. The dendrogram tree of 30K MPs and SJR CP hits obtained with MUSTANG in newick format.**
(NWK)

**S1 Fig. The per-residue confidence scores for AlphaFold2 (plDDT) and RoseTTAFold (Cα-lDDT) structural models.** Regions with lDDT > 90 are expected to be modeled to high accuracy, whereas regions with lDDT between 70 and 90 are expected to be modeled well (a generally good backbone prediction). Abbreviated virus names are explained in the legend of Fig 2. Numerical data used to generate the plDDT plots can be found in S3 Table.
(TIF)

**S2 Fig. The superimposition of the 30K MP of TMV (NP_597748) and the SJR CP from satellite tobacco mosaic virus (STMV, PDB: 1A34).**
(TIF)

**S3 Fig. Dendrogram and heatmap of complete linkage clustering of representative 30K MP and SJR CP based on the pairwise comparisons of the RMSD values calculated by MUSTANG.** The red circles indicated in the top dendrogram, represent bootstrap values ≥90 obtained with R package "pvclust." The CPs and MPs are indicated in blue and yellow, respectively.
(TIF)

**S4 Fig. Structural comparison of the pepino mosaic virus (PepMV) CP (PDB: 5FN1) and the putative MP of Charavirus canadiensis (QBG78689).** The structures are colored using the rainbow scheme from blue (N-terminus) to red (C-terminus) and α-helices equivalent between the 2 proteins are numbered. For the charavirus protein, only the region corresponding to the PepMV CP is shown.
(TIF)

**S5 Fig. The conservation of the D-motif in 30K MPs and SJR CPs.** The alignment was made using PROMALS3D. Only the region encompassing the D-motif is shown.
(TIF)

**S6 Fig. Plots of local charges in 21 amino acid sliding window for four 30K MP and four SJR CP representatives.** The jelly-roll region is marked in light green.
(TIF)

## Author Contributions

**Conceptualization:** Valerian V. Dolja, Eugene V. Koonin, Mart Krupovic.

**Formal analysis:** Anamarija Butkovic, Mart Krupovic.

**Investigation:** Anamarija Butkovic, Valerian V. Dolja, Eugene V. Koonin, Mart Krupovic.

**Supervision:** Mart Krupovic.

**Visualization:** Anamarija Butkovic.

**Writing – original draft:** Anamarija Butkovic, Eugene V. Koonin.

**Writing – review & editing:** Valerian V. Dolja, Mart Krupovic.

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
