## [Editor Report · Decision Letter 0]

12 Jan 2023

Dear Dr. Krupovic, 

Thank you for submitting your manuscript entitled "Origin of plant virus movement proteins from jelly-roll capsid proteins" for consideration as a Research Article by PLOS Biology.

Your manuscript has now been evaluated by the PLOS Biology editorial staff and I am writing to let you know that we would like to send your submission out for external peer review.

Once your full submission is complete, your paper will undergo a series of checks in preparation for peer review. After your manuscript has passed the checks it will be sent out for review. To provide the metadata for your submission, please Login to Editorial Manager (https://www.editorialmanager.com/pbiology) within two working days, i.e. by Jan 14 2023 11:59PM.

Kind regards,

Paula

---

Senior Editor

PLOS Biology

---

## [Decision Letter · Decision Letter 1]

24 Mar 2023

Dear Dr. Krupovic,

Please allow me to first apologize for the delay in the processing of your manuscript. This delay is caused by my difficulty in recruiting reviewers for your manuscript. I am sorry for this, and I thank you for your patience while your manuscript "Origin of plant virus movement proteins from jelly-roll capsid proteins" was peer-reviewed at PLOS Biology. It has now been evaluated by the PLOS Biology editors, an Academic Editor with relevant expertise, and by several independent reviewers. 

In light of the reviews, which you will find at the end of this email, we would like to invite you to revise the work to thoroughly address the reviewers' reports.

As you will see below, the reviewers find your work interesting but they raise several issues that should be solved before further consideration. In particular, we think it is very important that you show the confidence metrics for both AlphaFold and Rosetta, tone down statements as needed and reorganize the manuscript as needed. Both reviewers agree that you should be careful with your conclusion that geminivirus CP is an origin of MPs. Further, reviewer #2 considers that to draw a conclusion, some careful considerations on the predicted structures and the alignment are required, and this reviewer provides suggestions for data analysis to achieve this.

Given the extent of revision needed, we cannot make a decision about publication until we have seen the revised manuscript and your response to the reviewers' comments. Your revised manuscript is likely to be sent for further evaluation by all or a subset of the reviewers.

**IMPORTANT - SUBMITTING YOUR REVISION**

*Re-submission Checklist*

*Published Peer Review*

*PLOS Data Policy*

*Blot and Gel Data Policy*

Sincerely,

Paula

---

Senior Editor

PLOS Biology

REVIEWS:

Reviewer #1: Vitaly Citovsky. Plant viruses and movement proteins.

Reviewer #2: Structural virology and evolution.

Reviewer #1: The 30K family is the most prevalent among the movement proteins (MPs) of extremely diverse plant viruses that affect a broad variety of wild and crop plants alike. Despite the decades-long effort, however, the molecular mechanisms whereby 30K MPs empower plant virus infection remain poorly understood. In this respect, the work by Butkovic et al. represents a sea change by revealing 3D structures of a broad variety of 30K MPs. Methodologically, the authors used the most advanced bioinformatics available for the prediction and comparison of protein structures (AlphaFold2, RoseTTAFold, DALI), as well as a variety of more traditional, complementary approaches for the sequence and phylogenetic analyses. Furthermore, they generated a comprehensive database of 30K MPs that provides an important resource for the entire plant virology community. 

This work compellingly demonstrates the single jelly-roll (SJR) fold of the structural core of all 30K MP subfamilies. The authors also analyzed additional structural elements, such as N- and C-terminal core extensions in the 30K MPs of distinct subfamilies. Taken together, the findings of this study are certain to dramatically facilitate experimental analysis of the structure-to-function relationships within 30K MPs. 

Finally, this work reports a largely unexpected evolutionary discovery: the origin of the 30K MPs from the SJR capsid proteins of the icosahedral RNA and DNA viruses, representing an example of the functional repurposing of the virus proteins during expansion to a novel ecological niche.

What this reviewer is a bit less enthusiastic about, is the suggestion that the geminiviruses could be an original virus lineage in which SJR CP gene duplication and re-functionalization have occurred. From the studies of the global distribution of virus families, it appears that the host range and geography of geminiviruses were historically limited to mostly tropical areas. This distribution has dramatically expanded into temperate regions in the second part of the XX century, likely due to the expansion of the geographical range of the whiteflies (most common geminivirus vectors) along with global warming. Accordingly, it seems unlikely that the ssDNA geminiviruses which represent a relatively minor part of the plant virome were the original source for 30K MP emergence. Because the significant majority of plant virus families possessing diverse lineages of 30K MPs are RNA viruses, it seems more plausible that these MPs have emerged among the RNA viruses. 

Reviewer #2: Butkovic et al has studied structural similarity of a jelly-roll motif and evolutionary lineage between movement proteins (MPs) encoded in broadly RNA/DNA viruses (especially in plant viruses) and capsid proteins (CPs) in ssRNA and ssDNA viruses. The results largely depend on recently invented AI-based accurate structural predictions of structurally unrevealed MPs using AlphaFold2 and RoseTTAFold. Based on the authors' structure-based alignments' and similarity analysis, previously recognized conserved secondary structure and D-motif in MPs and CPs are deeply and comprehensively analyzed, which could explain their same evolutionary origin. Butkovic et al also hypothesizes likely evolutionary scenario of a MP gene acquisition from a CP gene.

The addressed question is very interesting in the sense of understanding function and origin of the 30K MPs superfamily in diverse viruses and the hypothesized evolutionary scenario is likely. Since the jelly-roll fold and D-motif of the MPs has already mentioned in previously papers, the main contribution of the manuscript has been the thoroughly analysis of the predicted MPs and CPs structures in diverse viruses. However, the main bottleneck is no experimental structure of the MP available for confirming the hypothesis and the results. Although I agree that Al-based structural prediction is a strong and useful tool for performing structure-based classifications, it should be cautiously of validating and interpreting the results owing not to being misled by the predicted models. Therefore, major and minor comments to be addressed.

Major and technical issues

1. Many parts of discussion and speculations are found in Results section and lengthy, and thus it is hard to read. I highly recommend reorganizing "Results" and "Discussion" or using an option of "Results and Discussion". The same or similar discussions are also found both in results and in discussion sections. It should be integrated to one.

2. Fig. 2A and lines 196-205. The most difficult part for me to review is that there is no available data for clarifying the accuracy of the MPs' structural prediction. AlphaFold2 automatically generates per-residue pLDDT score (e.g. Tunyasuvunakool et al., 2021, DOI: 10.1038/s41586-021-03828-1) to validate predicted and generated structures. The jelly-roll core of the MPs seems to be well predicted, however most likely loops, N-/C-terminal parts are not. Such validation data should be included in the manuscript and described in the main text. Also, it is mentioned that RoseTTAFold was used in case AlphaFold2 showed poor IDDT results, but then how to verify the RoseTTAFold models were accurate?

3. Fig. 3 and lines 206-238. It is not clear which part of the predicted structures are used for DALI calculation. If you used the entire predicted structures, the obtained DALI-score largely affected by badly predicted loops or N- and C-terminal regions. This is crucial for discussing the structural similarity in the MPs and the CPs. Could you generate a structure-based phylogeny using only accurately predicted jelly-roll domain of the MPs and the CPs? Also, DALI is a good start for identifying similar structures, but not suitable for generating accurate multiple structural alignments. There is a better alignment tool (e.g. HSF, SHP, MUSTANG...) to generate an RMSD-based phylogeny as seen in previously published papers (e.g. Riffel et al., 2002, DOI: 10.1016/S0969-2126(02)00896-1; Wang et al., 2014, DOI: 10.1038/nature13806). It should be very careful of concluding that geminivirus CP is an origin of MPs from the predicted structures.

4. Fig. 5B and lines 301-305. It is very risky to say a function of the D-motif in the predicted structure. For me, it is not clear why this amino acid residue on loop or turn between two beta-sheets is important for protein folding. Then, why amino acid residues in the other loops or turns in the jelly-roll fold are not conserved? Also, it is difficult to discuss about C-terminal and N-terminal structure, function, and evolution if the structural predictions in these parts are not accurate. As also mentioned several times in the main text, the predicted N-terminal and C-terminal structure is disordered/unstructured and I guess most likely because it failed to predict. I strongly recommend toning down results and discussion with regards to the C-terminal and the N-terminal.

Minor comments

1. Abstract, lines 29-31. As mentioned in a major comment, this part is too speculative. 

2. Line 227. It is not clear for me why MPs evolved from the CPs by considering their different distributions in species. MPs also seemed to find in algae viruses in this manuscript.

3. Line 238. (see above). It is not obvious what should be referred.

4. Lines 247-248. "...showed a perfect alignment...". I think it should not be perfect alignment, then two structures are the same structure.

5. Lines 317-319. RNA-binding domains of the MPs seem not to be concluded in my reading the referred old papers. I do not think it is a good idea of combining controversy data and predicted structural data.

6. Lines 322-325. This part is very speculative and no experimental data to support this idea.

7. Lines 335-341. These are all very speculative. Please tone down and make it much clear it is a speculation. Without resolving, e.g. complex structure of MPs, how to describe the impairment of the icosahedral capsid formation? 

8. Lines 362-386. I understand the point of the discussion regarding what likely happened to MPs after viruses adapting to plants and arthropods, however it is too lengthy and it is obscure how any of the findings in this manuscript support this hypothesis.

---

## [Decision Letter · Decision Letter 2]

3 May 2023

Dear Dr. Krupovic,

Thank you for your patience while we considered your revised manuscript "Origin of plant virus movement proteins from jelly-roll capsid proteins" for publication as a Research Article at PLOS Biology. This revised version of your manuscript has been evaluated by the PLOS Biology editors, the Academic Editor, and the original reviewers.

Based on the reviews, we are likely to accept this manuscript for publication, provided you satisfactorily address the following data and other policy-related requests.

1. DATA POLICY:

A) Supplementary files (e.g., excel). Please ensure that all data files are uploaded as 'Supporting Information' and are invariably referred to (in the manuscript, figure legends, and the Description field when uploading your files) using the following format verbatim: S1 Data, S2 Data, etc. Multiple panels of a single or even several figures can be included as multiple sheets in one excel file that is saved using exactly the following convention: S1_Data.xlsx (using an underscore).

B) Deposition in a publicly available repository. Please also provide the accession code or a reviewer link so that we may view your data before publication.

Regardless of the method selected, please ensure that you provide the individual numerical values that underlie the summary data displayed in the following figure panels as they are essential for readers to assess your analysis and to reproduce it: Figures 1AB, 3B, 5AC, and Supplementary Figures SF1, SF3, SF6.

**Please also ensure that figure legends in your manuscript include information on where the underlying data can be found, and ensure your supplemental data file/s has a legend.**

2. Please provide a blurb which (if accepted) will be included in our weekly and monthly Electronic Table of Contents, sent out to readers of PLOS Biology, and may be used to promote your article in social media. The blurb should be about 30-40 words long and is subject to editorial changes. It should, without exaggeration, entice people to read your manuscript. It should not be redundant with the title and should not contain acronyms or abbreviations.

3. We suggest a change in the title to make it more direct. Our suggestion: "Plant virus movement proteins originated from jelly-roll capsid proteins".

We expect to receive your revised manuscript within two weeks.

*Published Peer Review History*

*Press*

Sincerely,

Paula

---

Senior Editor,

pjaureguionieva@plos.org,

PLOS Biology

Reviewer remarks:

Reviewer #1: Vitaly Citovsky

Reviewer #2: Kenta Okamoto

Reviewer #1: the revised paper has addressed all the points raised in my previous review of this manuscript

Reviewer #2: The authors address all my comments and edit main text and figures to improve with regards to scientific intergrity and readability. Some of the predicted structures show a slightly low quality of the predictions, however the authors tone down the main text and show the validation data, which will not rule out other possibilities. Hence, the risk of overinterpreting the results is very minor. Therefore, I do not have any more comments. I am positive of the biological interests in the manuscript as mentioned in my former comment.

---

## [Editor Report · Decision Letter 3]

11 May 2023

Dear Dr. Krupovic,

Thank you for the submission of your revised Research Article "Plant virus movement proteins originated from jelly-roll capsid proteins" for publication in PLOS Biology. On behalf of my colleagues and the Academic Editor, Kenta Okamoto, I am pleased to say that we can in principle accept your manuscript for publication, provided you address any remaining formatting and reporting issues. These will be detailed in an email you should receive within 2-3 business days from our colleagues in the journal operations team; no action is required from you until then. Please note that we will not be able to formally accept your manuscript and schedule it for publication until you have completed any requested changes.

PRESS

Sincerely, 

Paula Jauregui

---

Senior Editor

PLOS Biology
